# Trends in immobilization following fracture surgery of the wrist and hand in Sweden: A 16-Year Analysis from 2008 to 2023

Michael Axenhus[1,2], Viktor Schmidt [1,2]*

1 Department of Orthopaedic Surgery, Danderyd Hospital, Stockholm, Sweden, 2 Department of Clinical Sciences at Danderyd Hospital, Karolinska Institutet, Stockholm, Sweden

* viktor.schmidt@ki.se

## Abstract

### Introduction

Hand and wrist fractures are common orthopaedic injuries, with varied postoperative management strategies. Cast immobilization, traditionally used to ensure fracture stability, is increasingly debated in favour of early mobilization. However, significant regional differences persist. The aim of this study is to analyse regional, age, and sex differences in post-operative cast immobilization rates for hand and wrist fractures in Sweden between 2008 and 2023.

### Methods

This observational, population-based study utilized data from the Swedish National Patient Register, analysing post-operative cast immobilization rates among individuals aged 15 and older. Trends and disparities were examined over time by region, gender, and age group.

### Results

The study found a significant increase in cast immobilization rates across Sweden, with notable differences based on sex and region. By 2023, immobilization rates for men had increased from 46% to 86%, while rates for women increased from 34% to 69%.

### Conclusion

The findings indicate a nationwide trend towards increased immobilization rates, highlighting gender- and region-based disparities. These differences underscore the need for standardized, evidence-based guidelines to ensure equitable and effective patient care across Sweden.

**Data availability statement:** The data used in this study is obtained from the website of the SNBHW and is publicly available for anyone to download and use. https://sdb.socialstyrelsen.se/if_ope/val.aspx

**Funding:** The author(s) received no specific funding for this work.

**Competing interests:** The authors have declared that no competing interests exist.

## Introduction

Hand and wrist fractures are among the most common fractures, accounting for a significant proportion of orthopaedic trauma worldwide and up to 30% of emergency department visits [1–3]. These fractures often result from low-energy falls in older adults with osteoporosis and high-energy trauma in younger individuals. Managing these injuries present challenges for effective management due to their association with an aging population and osteoporosis meanwhile also affecting younger patients who have high functional demands [4,5].

Over the years, the operative treatment of wrist fractures has evolved, shifting from external fixators and percutaneous pinning to volar locking plates, though without clear evidence of improved clinical outcomes [6]. Similarly, the management of hand fractures has progressed, with advancements in surgical techniques aimed at optimizing functional recovery [7]. Cast immobilization, traditionally employed to maintain fracture stability, is increasingly being reevaluated in favour of early mobilisation [8].

Postoperative (or nonoperative) management, including cast immobilization, remains a subject of debate. While discussions often focus on surgical techniques, the role of mobilization in recovery is crucial [9]. In Sweden, postoperative immobilization has historically been standard practice, particularly for fractures treated with k-wires. This has been questioned with the advent of more advanced fixation techniques and implants [10]. However, the lack of consensus regarding the necessity and duration of immobilization continues to influence clinical decision-making, both for wrist [11,12] and hand fractures [13]. Immobilization periods vary widely, from immediate mobilization to six weeks of casting [8,11,12,14,15], with significant regional, age and sex-related disparities.

Understanding regional variations in postoperative immobilization is essential for assessing equity and standardization in healthcare. Previous studies indicate that clinical practices often vary by region due to differences in resources, local guidelines, and patient demographics [16–18]. Moreover, sex disparities in treatment decisions remain an important consideration, as women—who constitute the majority of older fracture patients—may receive different treatment strategies compared to men. Age-related factors, including the increased prevalence of osteoporosis and decreased physiological reserve in older adults, further influence postoperative management choices. Recent national-level studies have identified regional differences in healthcare delivery, prompting research into this area [18,19].

The aim of this study is to analyse regional, sex, and age differences in postoperative cast immobilization practices to clarify trends and disparities that may inform future clinical guidelines and healthcare policies.

## Materials and methods

### Study design and setting

This observational, population-based analysis utilized open-access data on surgical procedures from the Swedish National Patient Register (NPR) for the period 2008–2023. The study follows the RECORD guidelines [20].

## Healthcare system context

Sweden's healthcare system provides universal access to medical services. Emergency care, inpatient treatment, and outpatient consultations are primarily offered in public hospitals, which serve the majority of patients with fractures. Each resident in Sweden has a unique personal identification number, enabling seamless data integration across national healthcare registers.

## Data source

The NPR is a comprehensive healthcare database containing inpatient and outpatient care records. It has maintained nationwide coverage since 1987 and includes detailed records on surgical interventions, including geographic information, patient demographics, and procedural specifics and has been validated for the use of epidemiological studies [21,22]. All hospitals in Sweden, both public and private, are mandated to report data to the NPR, ensuring comprehensive coverage. Diagnoses in the NPR are coded using the International Statistical Classification of Diseases, 10th Revision (ICD-10), and surgical procedures are coded following the NOMESCO classification system [23]. All orthopaedic and hand surgery departments across Sweden participate in the NPR.

## Study population

This analysis focuses exclusively on individuals aged 15 years and older who underwent surgery for hand and wrist fractures between January 1, 2008, and December 31, 2023. Eligible patients were identified using the NPR and included only if they possessed a valid Swedish personal identification number, ensuring linkage across healthcare registers. Procedures were identified using NOMESCO surgical code TND32 for peroperative cast immobilisation and and NDJ for surgical fixation of fractures. The number of individuals with these procedures were recorded and analysed.

## Variables

The study variables include patient age, sex, and geographic location. Sex is classified as male or female, and regional data were extracted to evaluate differences in treatment practices across Sweden.

## Statistics

Annual incidence rates (per 100,000 person-years)), of patients having both a TND32 and NDJ code, were determined using population data for people aged ≥15 years in each Swedish healthcare region, sourced from Statistics Sweden. Confidence intervals (CIs) for incidence rates were estimated using Poisson regression with the exact method. We used multivariate Poisson regression analyses to explore regional variation in incidence rates and associated factors: Relationship between current, 2023, and historical, 2008, incidences, controlling for regional population sizes. For consistency in the models, regional population size was expressed per 100,000 inhabitants. Results were reported as incidence with corresponding 95% CIs where appropriate. Model residuals were validated for normal distribution using Q–Q plots and the Shapiro–Wilk test. All statistical analyses were conducted using R software (version 4.3.2, R Foundation for Statistical Computing, Vienna, Austria). Regional maps graphs were created in Illustrator 2025 (29.5.1).

## Ethics

The study was performed using open-source data and was therefore not subject to ethical review.

## Results

In 2008, approximately 40% of all surgically treated hand and wrist fractures received postoperative cast immobilization. Men had a higher immobilization rate (46%) compared to women (34%). By 2023, immobilization rates had increased significantly, reaching 86% for men and 69% for women (Fig 1, Table 1).

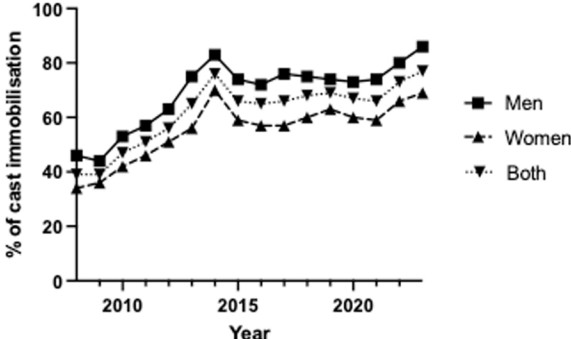

**Fig 1. Percentage of cast immobilisation following fracture surgery for hand or wrist fracture during 2008-2023.**

**Table 1. Percentage of post-operative cast immobilization for hand or wrist fractures following surgical fixation across Swedish regions in 2008 and 2023, with change over time\*.**

| | 2008 | | | 2023 | | | Change men | Change women | Change both |
|---|---|---|---|---|---|---|---|---|---|
| Region | Men | Women | Both | Men | Women | Both | 2008–2023 | 2008–2023 | 2008–2023 |
| Sweden | 46% | 34% | 40% | 86% | 69% | 78% | 41% | 35% | 38% |
| Stockholm | 16% | 14% | 15% | 42% | 41% | 42% | 26% | 26% | 27% |
| Uppsala | 8% | 4% | 6% | 4% | 18% | 11% | −4% | 14% | 5% |
| Södermanland | 14% | 15% | 15% | 99% | 52% | 76% | 85% | 37% | 61% |
| Östergötland | 31% | 28% | 30% | 98% | 99% | 99% | 67% | 71% | 69% |
| Jönköping | 13% | 15% | 14% | 99% | 46% | 73% | 86% | 31% | 59% |
| Kronoberg | 0% | 14% | 7% | 26% | 19% | 23% | 26% | 5% | 16% |
| Kalmar | 31% | 54% | 43% | 96% | 96% | 96% | 65% | 42% | 54% |
| Gotland | 63% | 60% | 62% | 75% | 74% | 75% | 12% | 14% | 13% |
| Blekinge | 10% | 99% | 55% | 97% | 99% | 98% | 87% | 0% | 44% |
| Skåne | 24% | 25% | 25% | 45% | 73% | 59% | 21% | 48% | 35% |
| Halland | 59% | 32% | 46% | 99% | 97% | 98% | 40% | 65% | 53% |
| Västra Götaland | 96% | 87% | 92% | 98% | 99% | 99% | 2% | 12% | 7% |
| Värmland | 19% | 14% | 17% | 45% | 60% | 53% | 26% | 46% | 36% |
| Örebro | 15% | 18% | 17% | 15% | 23% | 19% | 0% | 5% | 3% |
| Västmanland | 7% | 7% | 7% | 36% | 57% | 47% | 29% | 50% | 40% |
| Dalarna | 16% | 20% | 18% | 24% | 13% | 19% | 8% | −7% | 1% |
| Gävleborg | 17% | 10% | 14% | 44% | 30% | 37% | 28% | 20% | 24% |
| Västernorrland | 41% | 6% | 24% | 54% | 65% | 60% | 13% | 58% | 36% |
| Jämtland | 3% | 8% | 6% | 7% | 33% | 20% | 4% | 26% | 15% |
| Västerbotten | 78% | 38% | 58% | 74% | 98% | 86% | −4% | 60% | 28% |
| Norrbotten | 82% | 63% | 73% | 59% | 67% | 63% | −23% | 3% | −10% |

\*Change over time in absolute percentage points

Marked regional differences were observed (Tables 1 and 2). For instance, Stockholm saw increases for both men (16% to 42%) and women (14% to 41%), while Uppsala had a decrease in men's cast usage (8% to 4%) and an increase for women (4% to 18%) (Table 1).

**Table 2.** Total number of patients treated with cast after surgical treatment of the wrist or hand in Swedish regions in 2008 and 2023, with percentage change over this period.

| Region | 2008 | 2023 | Change 2008/2023 |
|---|---|---|---|
| Sweden | 8 589 | 10 221 | 119% |
| Stockholm | 1 824 | 2 046 | 112% |
| Uppsala | 345 | 272 | 79% |
| Södermanland | 209 | 288 | 138% |
| Östergötland | 316 | 615 | 195% |
| Jönköping | 412 | 446 | 108% |
| Kronoberg | 123 | 234 | 190% |
| Kalmar | 154 | 191 | 124% |
| Gotland | 31 | 42 | 135% |
| Blekinge | 77 | 101 | 131% |
| Skåne | 1 053 | 1 511 | 143% |
| Halland | 215 | 298 | 139% |
| Västra Götaland | 1 459 | 1 655 | 113% |
| Värmland | 312 | 209 | 67% |
| Örebro | 263 | 324 | 123% |
| Västmanland | 311 | 418 | 134% |
| Dalarna | 417 | 247 | 59% |
| Gävleborg | 170 | 297 | 175% |
| Västernorrland | 150 | 435 | 290% |
| Jämtland | 114 | 123 | 108% |
| Västerbotten | 308 | 235 | 76% |
| Norrbotten | 326 | 234 | 72% |

There was an increased incidence of surgically treated wrist and hand fractures during the study period (Fig 2). However, when comparing women and men, the incidence among women increased, while it decreased among men over the same timeframe. Moreover, there was a notable decrease in the incidence of surgeries in 2020, likely due to the impact of the COVID-19 pandemic.

Nationally, the use of post-operative cast immobilization increased by almost 100%, from 40% in 2008 to 78% in 2023 (Fig 1). However, regional differences are notable. Some areas, like Norrbotten, saw a decrease of 14%, while regions like Södermanland and Västmanland experienced large increases of 407 and 586%, respectively. Regions such as Dalarna and Västra Götaland did not see big changes, with increases of about 5% (Table 1).

Analyzing sex, there are even greater regional differences. For instance, Stockholm saw increases for both men (16% to 42%) and women (14% to 41%). Meanwhile, Uppsala had a decrease in men's cast usage (8% to 4%) and an increase for women (4% to 18%), and Dalarna had an increase in men's cast usage (16% to 24%) and a decrease for women (20% to 13%) (Table 1).

The percentage of post-operative cast immobilization for hand and wrist fractures generally increases with age, with the highest numbers observed in the oldest age group (85+), reaching values nearing 100% for both men and women. The 80–84 age group also shows notably high percentages, while younger groups, have more moderate and fluctuating rates (Fig 3).

From 2008 to 2023, the incidence of post-operative cast immobilization for fractures of the wrist and hand varied significantly across Swedish Regions and between sexes. Regions like Uppsala and Örebro show low immobilization rates, around 10%, with minimal gender differences. In contrast, Västra Götaland and Östergötland display the highest rates, reaching almost 100% (Fig 4).

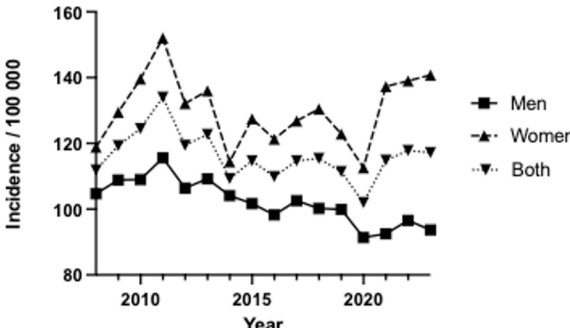

**Fig 2. Incidence of surgically treated wrist and hand fractures during 2008 to 2023.**

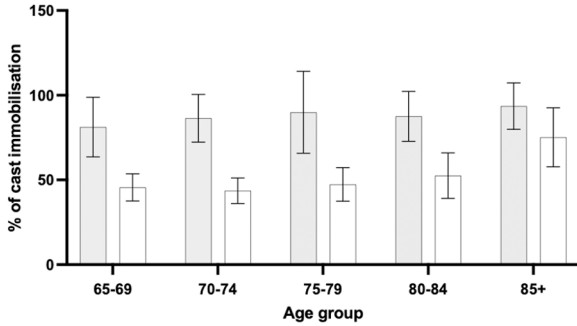

**Fig 3. Percentage of post-operative cast immobilisation between age groups.** Men indicated by grey bars; women indicated by white bars. Error bars indicate 95% confidence interval.

In 2008, higher immobilization rates are particularly evident in northern regions (with the exception of Västra Götaland), where percentages are close to 100%. By 2023, rates seem to generally equalize across the country (Fig 5).

## Discussion

This study highlights a substantial increase in postoperative cast immobilization rates over 16 years for hand and wrist fractures, with notable disparities across gender and geographic regions.

Sex differences were consistent throughout the study period, with men experiencing higher immobilization rates than women. This may be due to differences in fracture type, clinical decision-making, patient preferences, or injury severity, as men are more prone to be involved in high energy trauma [24]. Since we have unsorted data on both hand and wrist fractures, it is hard to discern what the main contributing factors are.

Significant regional differences in immobilization rates were observed, indicating variability in clinical practices across Sweden. Some regions, such as Södermanland and Västmanland, experienced dramatic increases in immobilization rates—407% and 586%, respectively. In contrast, regions like Norrbotten saw a decrease in immobilization rates over the study period.

These regional disparities suggest that local clinical guidelines, available resources, and clinician preferences heavily influence immobilization practices. Regional disparities in healthcare have been described before in Sweden [16,17]. However, this is to our knowledge the first nation-wide study that shown regional disparities in cast immobilization. Regional differences in immobilization practices may stem in part from variability in surgeon training and practice settings. Surgeons who trained at high-volume academic centers or completed specialized hand fellowships might adopt more progressive,

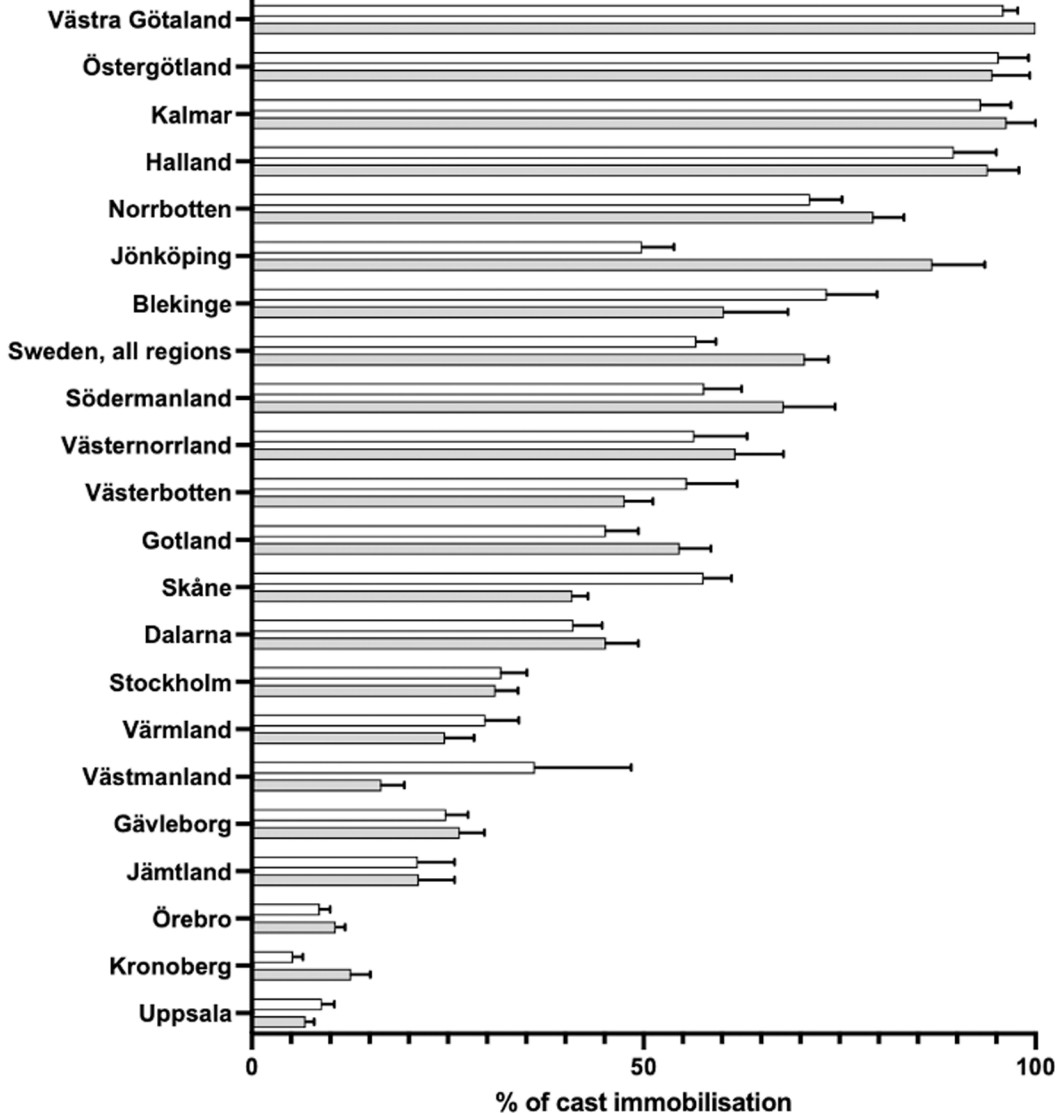

**Fig 4. Percentage of post-operative cast immobilization following hand and wrist fractures in Sweden, 2008–2023, by region and gender.** Grey bars represent men, white bars represent women. Error bars indicate 95% confidence intervals.

evidence-based postoperative protocols, including earlier mobilization. By contrast, those practicing in smaller community or rural hospitals—often with more generalized training—may adhere to traditional conservative approaches (e.g., routine casting for several weeks). Indeed, the lack of universally endorsed protocols means postoperative care often reflects the surgeon's personal training and preferences [25]. A recent review highlighted that "where an orthopedic surgeon trained, whether they obtained a hand or upper extremity fellowship, and their current practice location are all potential influencing factors" in immobilization strategy [25]. Thus, if one Swedish region's orthopedic staff has a strong academic or subspecialty background, they may be more comfortable with limited immobilization and early motion.

This phenomenon is not unique to Sweden. In the United States, for example, practice variation in hand surgery aftercare is well documented despite similar training standards nationally. One survey of hand surgeons found large variations

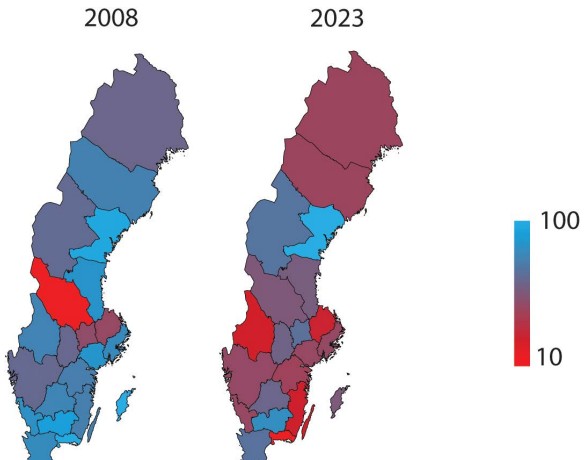

**Fig 5. Geographic distribution of percentage of for post-operative cast immobilization following hand or wrist fractures across Swedish Regions in 2008 and 2023.** The color gradient represents the percentage of immobilisation following surgery for hand or wrist fracture, with darker shades indicating higher incidence rates.

in the duration of postoperative immobilization for common procedures and only minimal correlation with years of experience [26]. The authors attributed this variability to the absence of high-level evidence guiding immobilization, leading surgeons to rely on their training and "personal preferences" [26]. These findings underscore how differences in training environments and mentorship can imprint lasting preferences on surgeons, contributing to regional disparities when those surgeons practice in different areas.

Until the advent of Swedish national guidelines in 2021, there was no single consensus to override these local habits. Notably, differences both between and within countries in managing distal radius fractures have been reported, leading several nations to publish guidelines to reduce unwarranted variation [27]. Sweden's new guidelines were developed to introduce "fast-track" criteria for early surgery and, implicitly, more uniform postoperative care [27]. Over time, such national guidance and increased dissemination of evidence may gradually harmonize immobilization practices. But in the period studied, regional medical culture—shaped by long-standing local guidelines (or lack thereof) and the attitudes of senior clinicians—likely drove much of the observed variation.

Age seems to be a significant factor in determining immobilization practices, with older patients experiencing higher rates of postoperative immobilization. The highest immobilization rates were observed in the oldest age groups, particularly among those aged 85 and older. This likely reflects the increased fracture fragility and complexity in older adults, where immobilization provides additional protection during the healing process. Clinicians may opt for immobilization in older patients to minimize the risk of complications and ensure optimal healing, considering factors such as comorbidities and decreased physiological reserve.

The observed increase in immobilization rates even among younger populations suggests a broader trend towards post-operative casting across all age groups. The reasons for the observed increase are speculative. One possible explanation is the presence of less experienced surgeons who may lack the confidence or expertise to initiate early mobilization following surgery.

The findings of this study emphasize the need for standardized clinical guidelines to reduce regional and demographic disparities in postoperative immobilization practices for hand and wrist fractures. This is in line with other studies which have shown the lack of standardized immobilization protocols [26,28]. There is currently insufficient evidence to definitively suggest cast immobilization following fracture surgery for some wrists fractures [29,30]. Some

studies also indicate that immediate mobilization might have functional benefits over immobilization, raising questions regarding the most suited clinical approach [31,32]. The substantial variability in immobilization rates suggests that patient care may be influenced by factors unrelated to individual patient needs, potentially affecting outcomes. Developing evidence-based protocols that consider the latest research on immobilization versus early mobilization could help ensure more consistent patient outcomes across regions and demographic groups. Randomized controlled trials are currently underway in order to determine the correct immobilization protocols following distal radius fractures but more trials are needed to determine the efficacy of immobilization following other hand injuries [9]. Standardized guidelines would assist clinicians in making informed decisions based on best practices, reducing the influence of regional practices or individual clinician preferences.

Further research is needed to understand the impact of postoperative immobilization on patient outcomes. Studies comparing the efficacy of immobilization versus early mobilization in terms of functional recovery, complication rates, and patient satisfaction would provide valuable insights. Additionally, the data in this study lacks the specificity needed to draw definitive conclusions and primarily highlights emerging trends. More granular data is required to thoroughly explore this topic. An essential next step would be a national, retrospective cohort study focusing on specific surgical treatments and postoperative protocols for particular diagnoses—for example, examining only volar plating for distal radius fractures and its associated postoperative management.

The study has several limitations. While the Swedish NPR provides comprehensive data, potential inaccuracies in coding and reporting may affect results. The results of this study could in part be explained by a change in registration of cast fixation after surgical treatment of these fractures. The study does not account for fracture severity, specific fracture types, detailed fracture location, surgical techniques or patient comorbidities, all of which can influence immobilization decisions. These are critical confounders that limit causal inference about why immobilization trends differed by region. For example, a region treating a higher proportion of complex or unstable fractures (e.g., comminuted intra-articular injuries) would appropriately use longer immobilization on average, independent of surgeon preference.

Moreover, regional differences in resources and clinical practices are inferred rather than directly measured. Our findings are consistent with that understanding, but the inability to adjust for severity or technique means we cannot definitively claim that training, infrastructure, or culture caused the immobilization differences – only that they are plausible contributing factors. This limitation highlights a need for more detailed, prospective data to confirm the reasons behind regional practice patterns.

## Conclusion

This study reveals significant variability in post-operative cast immobilization practices for hand and wrist fractures in Sweden from 2008 to 2023, with higher rates among men and older patients, and notable regional differences which could be influenced by local guidelines and resources. The overall trend towards increased immobilization highlights a shift in clinical practice over time. These findings underscore the need for standardized, evidence-based clinical protocols to ensure equitable, evidence-bases care and optimize patient outcomes. Larger studies that address the long-term outcomes of immobilization versus mobilization are essential to guide stakeholders in implementing effective changes in postoperative management.

## Acknowledgments

Not applicable

## Author contributions

**Conceptualization:** Viktor Schmidt, Michael Axenhus.

**Data curation:** Michael Axenhus.

Formal analysis: Michael Axenhus.

Funding acquisition: Michael Axenhus.

Investigation: Viktor Schmidt, Michael Axenhus.

Methodology: Viktor Schmidt, Michael Axenhus.

Project administration: Viktor Schmidt, Michael Axenhus.

Resources: Michael Axenhus.

Software: Michael Axenhus.

Supervision: Viktor Schmidt, Michael Axenhus.

Validation: Viktor Schmidt, Michael Axenhus.

Visualization: Viktor Schmidt, Michael Axenhus.

Writing – original draft: Viktor Schmidt, Michael Axenhus.

Writing – review & editing: Viktor Schmidt.

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
