## [Decision Letter · Decision Letter 0]

PONE-D-25-05927Trends in immobilisation following fracture surgery of the wrist and hand in Sweden: A 16-Year Analysis from 2008 to 2023PLOS ONE

Dear Dr. Schmidt,

Thank you for submitting your manuscript to PLOS ONE. After careful consideration, we feel that it has merit but does not fully meet PLOS ONE’s publication criteria as it currently stands. Therefore, we invite you to submit a revised version of the manuscript that addresses the points raised during the review process.

Expand the discussion on potential drivers of regional differences (e.g., surgeon training, resource availability) beyond speculative explanations.

Acknowledge that the lack of fracture severity/surgical technique data limits causal interpretations of immobilization trends.

We look forward to receiving your revised manuscript.

Kind regards,

Xiaoen Wei

Academic Editor

PLOS ONE

Additional Editor Comments:

Expand the discussion on potential drivers of regional differences (e.g., surgeon training, resource availability) beyond speculative explanations.

Acknowledge that the lack of fracture severity/surgical technique data limits causal interpretations of immobilization trends.

Reviewers' comments:

Reviewer's Responses to Questions

**Comments to the Author**

1. Is the manuscript technically sound, and do the data support the conclusions?

Reviewer #1: Yes

2. Has the statistical analysis been performed appropriately and rigorously? 

Reviewer #1: No

3. Have the authors made all data underlying the findings in their manuscript fully available?

Reviewer #1: Yes

4. Is the manuscript presented in an intelligible fashion and written in standard English?

Reviewer #1: Yes

5. Review Comments to the Author

Reviewer #1: Expand the discussion on potential drivers of regional differences (e.g., surgeon training, resource availability) beyond speculative explanations.

Acknowledge that the lack of fracture severity/surgical technique data limits causal interpretations of immobilization trends.

6. PLOS authors have the option to publish the peer review history of their article (what does this mean? ). If published, this will include your full peer review and any attached files.

**Do you want your identity to be public for this peer review?** For information about this choice, including consent withdrawal, please see our Privacy Policy .

Reviewer #1: No

---

## [Editor Report · Decision Letter 1]

Trends in immobilization following fracture surgery of the wrist and hand in Sweden: A 16-Year Analysis from 2008 to 2023

PONE-D-25-05927R1

Dear Dr. Schmidt,

We’re pleased to inform you that your manuscript has been judged scientifically suitable for publication and will be formally accepted for publication once it meets all outstanding technical requirements.

Kind regards,

Xiaoen Wei

Academic Editor

PLOS ONE
---

## [Editor Report · Acceptance letter]

PONE-D-25-05927R1

PLOS ONE

Dear Dr. Schmidt,

I'm pleased to inform you that your manuscript has been deemed suitable for publication in PLOS ONE. Congratulations! Your manuscript is now being handed over to our production team.

Kind regards,

on behalf of

Dr. Xiaoen Wei

Academic Editor

PLOS ONE